# Molecular Hydrogen Enhances Proliferation of Cancer Cells That Exhibit Potent Mitochondrial Unfolded Protein Response

**DOI:** 10.3390/ijms23052888

**Published:** 2022-03-07

**Authors:** Tomoya Hasegawa, Mikako Ito, Satoru Hasegawa, Masaki Teranishi, Koki Takeda, Shuto Negishi, Hiroshi Nishiwaki, Jun-ichi Takeda, Tyler W. LeBaron, Kinji Ohno

**Affiliations:** 1Division of Neurogenetics, Center for Neurological Diseases and Cancer, Nagoya University Graduate School of Medicine, Nagoya 466-8550, Japan; hasegawa.tomoya@med.nagoya-u.ac.jp (T.H.); ito@med.nagoya-u.ac.jp (M.I.); hasegawa.satoru0530@gmail.com (S.H.); teranishi.masaki.u5@s.mail.nagoya-u.ac.jp (M.T.); mhf.07236174@gmail.com (K.T.); negishi.shuto@med.nagoya-u.ac.jp (S.N.); nishiwaki.h@med.nagoya-u.ac.jp (H.N.); jtakeda@med.nagoya-u.ac.jp (J.-i.T.); 2Molecular Hydrogen Institute, Enoch City, UT 84721, USA; sci_ty7@yahoo.com; 3Centre of Experimental Medicine, Institute for Heart Research, Slovak Academy of Sciences, 84104 Bratislava, Slovakia; 4Department of Kinesiology and Outdoor Recreation, Southern Utah University, Cedar City, UT 84720, USA

**Keywords:** molecular hydrogen, cellular proliferation, cancer cell lines, mitochondrial unfolded protein response, mitochondrial electron transfer chain

## Abstract

Molecular hydrogen ameliorates pathological states in a variety of human diseases, animal models, and cell models, but the effects of hydrogen on cancer have been rarely reported. In addition, the molecular mechanisms underlying the effects of hydrogen remain mostly unelucidated. We found that hydrogen enhances proliferation of four out of seven human cancer cell lines (the responders). The proliferation-promoting effects were not correlated with basal levels of cellular reactive oxygen species. Expression profiling of the seven cells showed that the responders have higher gene expression of mitochondrial electron transport chain (ETC) molecules than the non-responders. In addition, the responders have higher mitochondrial mass, higher mitochondrial superoxide, higher mitochondrial membrane potential, and higher mitochondrial spare respiratory capacity than the non-responders. In the responders, hydrogen provoked mitochondrial unfolded protein response (mtUPR). Suppression of cell proliferation by rotenone, an inhibitor of mitochondrial ETC complex I, was rescued by hydrogen in the responders. Hydrogen triggers mtUPR and induces cell proliferation in cancer cells that have high basal and spare mitochondrial ETC activities.

## 1. Introduction

Molecular hydrogen (H_2_) can readily access any organ and cellular organelle due to its small molecular size and non-polar nature [1]. Some prokaryotes carry hydrogenases to metabolize hydrogen, and some rare non-mammalian eukaryotes carry hydrogenases [2,3]. Although hydrogenases do not exist in higher eukaryotes, hydrogen exerts disease-ameliorating effects in humans and rodents [4,5,6]. The effects of hydrogen have been repeatedly reported especially in oxidative stress, inflammatory diseases, and metabolic diseases [6]. Hydrogen was first reported to ameliorate cerebral infarction, ostensibly due to hydrogen’s ability to reduce hydroxyl radicals and peroxynitrite [7]. However, subsequent studies revealed a variety of signal-modulating activities, which cannot be readily accounted for by radical scavenging activities. For example, hydrogen activates the Nrf2/Keap1 signaling pathway, which, contrary to radical scavenging, is usually activated by oxidative stress [8,9,10,11,12,13,14,15]. We reported that hydrogen suppresses abnormally activated Wnt/β-catenin signaling by enhancing the activity of β-catenin degradation complex, and hydrogen ameliorates a mouse model of osteoarthritis by suppressing Wnt/β-catenin signaling [16]. We also reported, by gene expression profilings, that hydrogen induces heat shock response, which subsequently upregulates collagen biosynthesis and downregulates cell cycle [17]. Heat shock protein 60 (HSP60) and other mitochondrial unfolded protein response (mtUPR)-related molecules are induced in mice after drinking hydrogen-rich water [18]. mtUPR is induced in response to stress to maintain mitochondrial proteostasis [19]. Accumulation of unfolded and misfolded proteins in mitochondria induces phosphorylation of the eukaryotic translation initiation factor 2α (eIf2α), which is mediated by kinases including protein kinase R (PKR) and eukaryotic translation initiation factor 2-alpha kinase 4 (EIF2AK4) [20]. Phosphorylation of eIf2α has dual effects on protein translation. It induces extensive downregulation of protein translation and promotes the translation of activating transcription factors (ATF) 4 and 5, and C/EBP homologous protein (CHOP) [21]. CHOP activates the transcription of chaperones including HSP60 and mitochondrial homolog of Lon protease 1 (LonP1) [22] to eliminate misfolded proteins. This mitochondrion-nucleus crosstalk works to relieve oxidative stress for cell survival [23] and it also promotes cancer cell proliferation [24].

We observed that hydrogen facilitated the proliferation of four cancer cell lines (the responders), but not of three cancer cell lines (the non-responders). We dissected the molecular mechanisms that determined the different proliferation profiles between the two groups of cancer cell lines. We found that the responders had higher mitochondrial gene expression and higher mitochondrial electron transport chain (ETC) activities than the non-responders. Moreover, hydrogen induced mtUPR only in the responders.

## 2. Results

### 2.1. Hydrogen Promotes Cell Proliferation in Four of Seven Cancer Cell Lines

In order to examine the effects of molecular hydrogen on cancer cell growth, the proliferation of seven human cancer cell lines (A549, HCT116, HeLa, HepG2, HT1080, PC3, and SH-SY5Y cells) was analyzed by a bromodeoxyuridine (BrdU) assay after incubation for 48 h under 5% and 10% hydrogen gas, or 10% nitrogen gas. As seen in Figure 1, the proliferation of four cell lines (A549, HeLa, HT1080, and PC3 cells) was increased 1.16–1.27-fold by 5% hydrogen gas, and 1.30–1.41-fold by 10% hydrogen gas compared to 10% nitrogen gas. On the other hand, the proliferation of the other three cell lines (HCT116, HepG2, and SH-SY5Y cells) was not significantly increased by either 5% (0.97–1.06-fold) or 10% (0.99–1.05-fold) hydrogen gas.

### 2.2. Effects of Hydrogen on Cell Proliferation Are Independent of Concentrations of Cellular Reactive Oxygen Species (ROS)

To examine whether the difference in the cellular responses to hydrogen was accounted for by the difference in the reduction of ROS, we evaluated the concentrations of cellular ROS by DCFDA at the basal level, and after incubation under 10% hydrogen gas for 6 and 24 h. Basal ROS levels of the responders and non-responders were statistically similar on average (Figure 2). Similarly, incubation of the seven cell lines under 10% hydrogen gas for 6 and 24 h variably changed cellular ROS levels from cell line to cell line, and similarly reduced the cellular ROS levels in both the responders and non-responders on average (Appendix A). Therefore, the differences in the effects of hydrogen on ROS levels were unlikely to account for the differences in the proliferative responses to hydrogen in the seven cell lines.

### 2.3. High Expression of Mitochondrial Genes Predicts Hydrogen-Mediated Cell Proliferation

We next analyzed gene expression profiles of the responders and non-responders. For each of the seven cancer cell lines, four representative datasets of expression microarray of untreated cells were obtained from the GEO database. The expression profiles were then analyzed by gene set enrichment analysis (GSEA). Nineteen gene sets were expressed more in the responders than in the non-responders (familywise error rate, FWER, *p*-value < 0.05) (Figure 3a), whereas four gene sets were expressed more in the non-responders (Figure 3b). Out of the 19 highly expressed gene sets in the responders, the 12 most expressed gene sets were related to mitochondrial ETC activities (closed bars in Figure 3a). Among them, the top gene set was “ATP synthesis coupled electron transport” (Appendix A), and the second gene set was “respiratory chain activity”. In contrast, no features were shared in the four gene sets that were expressed more in the non-responders.

We also conducted whole-exome sequencing analysis of each cell line to search for single nucleotide variants (SNVs) that were shared either in the responders or the non-responders. The sequencing data were deposited at DDBJ with an accession number DRA011366. We found that no SNVs were shared in two or more cell lines in either group.

### 2.4. The Responders Exhibit Higher Mitochondrial ETC Activities and Shorter Doubling Time Than the Non-Responders

We next evaluated mitochondrial mass, mitochondrial superoxide, and mitochondrial membrane potential without hydrogen in the seven cell lines. Flow cytometric analysis showed that mitochondrial mass was higher in the responders than in the non-responders (Figure 4a). Similarly, mitochondrial superoxide (Figure 4b) and mitochondrial membrane potential (Figure 4c) were higher in the responders. In contrast, temporal profiles of mitochondrial superoxide (Appendix A) and mitochondrial membrane potentials (Appendix A) after exposure to hydrogen were similar between HT1080 cells (a responder) and HepG2 cells (a non-responder).

We also measured mitochondrial oxygen consumption rate (OCR), an indicator of mitochondrial ETC activities, in the absence of hydrogen in the seven cell lines using an extracellular flux analyzer (Figure 4d). Mitochondrial OCRs of the seven cell lines were normalized at the basal level so that the values before adding oligomycin became 1.0 (Phase 1 in Figure 4d). Normalized proton leaks (Phase 2 in Figure 4d), as well as normalized mitochondria-independent oxygen consumptions (Phase 4 in Figure 4d), were similar in all seven cell lines. However, maximum mitochondrial oxygen consumption after adding an uncoupling reagent, FCCP, was higher in the responders than in the non-responders (Phase 3 in Figure 4d). Similarly, relative spare respiratory capacities, which were calculated by (Phase 3 − Phase 4)/(Phase 1 − Phase 4) − 1, were higher in the responders (Figure 4e). Thus, the responders had more spare capacities to augment mitochondrial ETC activities.

The doubling time of the seven cell lines without hydrogen was calculated using the temporal profile of cell confluency. The responders had a shorter doubling time than the non-responders (Supplemental Appendix A). Similarly, the doubling time tended to be inversely correlated with the mitochondrial membrane potential (*r* = −0.667, *p* = 0.102), which was in accordance with previous reports [25,26,27,28].

### 2.5. Molecular Hydrogen Induces Mitochondrial Unfolded Protein Response (mtUPR) in Hydrogen-Responsive Cells

High mitochondrial gene expression and high mitochondrial ETC activities in the responders prompted us to hypothesize that mitochondria-related signals are readily induced by hydrogen in the responders. We thus examined the expression of mtUPR-related molecules in the seven cell lines. We found that 10% hydrogen gas induced the expression of mtUPR-related molecules in the responders (Figure 5). HSP60 was increased 1.35–1.87-fold in the responders (Figure 5a,b). Its upstream molecule in mtUPR, p-eIf2α was also increased on average in the responders, but not in the non-responders (Figure 5c,d). Similarly, a further upstream molecule in the mtUPR signaling pathway, ATF5, was also increased on average in the responders, but not in the non-responders (Figure 5c–e). Thus, hydrogen induces mtUPR as evidenced by upregulation of mtUPR-related molecules (ATF5, p-eIf2α, and HSP60) in the responders.

### 2.6. Molecular Hydrogen Modulates Mitochondrial Stress and Promotes Cell Proliferation

As mtUPR is critical for relieving mitochondrial stress, we examined whether hydrogen alleviates mitochondrial stress provoked by rotenone, an inhibitor of mitochondrial ETC complex I. Suppression of cell proliferation by rotenone at LD20 was partly rescued by hydrogen in a dose-dependent manner in the responders but not in the non-responders (Figure 6). Thus, rotenone suppressed cell proliferation in both the responders and non-responders by attenuating mitochondrial energy production, and mtUPR induced only in the responders rejuvenated mitochondrial energy production and rescued the suppressed cell proliferation.

## 3. Discussion

The beneficial effects of hydrogen have been reported in a plethora of human diseases and animal models [6]. We showed that molecular hydrogen provokes mtUPR and enhances cell proliferation in four out of the seven tested human cancer lines. The four hydrogen-responsive cells have higher expression of mitochondrial ETC genes and higher mitochondrial ETC activities than the three hydrogen-non-responsive cells.

We looked into single nucleotide variants (SNVs) in the seven cancer cell lines, which may be responsible for differential expression of mitochondrial ETC genes between the responders and the non-responders by exome-seq analysis. However, we could not find any responsible SNVs that were shared in either group. The determinants may be in copy-number variations (CNVs) or epigenetic expression regulation. Alternatively, mitochondrial ETC gene expression may be regulated by multiple genes, and no shared SNVs or CNVs may exist in either group.

The mitochondrial ETC is composed of four protein complexes, whose genes are encoded in mitochondrial DNA as well as genomic DNA. In our GSEA analysis, both mitochondrial genes and nuclear-encoded mitochondrial genes were enriched in the responders. Mitochondrial ETC accepts an electron from NADH or FADH to eventually pass it to oxygen. In the course of proton-coupled electron transfer through mitochondrial ETC complexes, a membrane potential is generated, and superoxide is inevitably produced. We found that the responders had larger mitochondrial mass (Figure 4a), as well as higher basal (Figure 4b,c) and spare (Figure 4d,e) ETC activities than the non-responders. The total cellular ROS levels measured by DCFDA were similar between the responders and the non-responders (Figure 2). In contrast, mitochondrial superoxide levels measured by MitoSOX were markedly higher in the responders (Figure 4b). This apparent discrepancy may be reconciled by considering the following. DCFDA reacts with H_2_O_2_, peroxynitrite, hydroxyl radicals, and to a lesser extent with superoxide. However, DCFDA cannot go through the mitochondrial inner membrane, so it cannot readily react with mitochondrial superoxide. In contrast, MitoSOX reacts with mitochondrial superoxide, but its entry into mitochondrial matrix is dependent on mitochondrial membrane potential. The high MitoSOX signals in the responders may represent high mitochondrial membrane potentials, as we observed with TMRM (Figure 4c). Thus, the basal mitochondrial ROS levels are similar between the responders and the non-responders.

Mitohormesis represents a paradoxically favorable response to noxious stimuli to mitochondria. mtUPR is a recently identified mediator of mitohormesis and induces cellular proliferation [23] and cellular survival [29]. Oxidative and other stresses trigger mtUPR to transmit a signal from mitochondria to the nucleus to maintain mitochondrial proteostasis [30]. Indeed, mtUPR and subsequent induction of ATF5 relieve ischemia-reperfusion injury in the heart [31]. Similarly, hydrogen induces the expression of ATF4 and phosphorylation of eIf2α in a rat model of cardiac infarction [32]. mtUPR-mediated phosphorylation of eIf2α decreases protein translation, which culminates in temporal reduction of oxidative phosphorylation activities by ETC [33]. Indeed, the lack of HSP60 [34] and another mtUPR-related molecule, caseinolytic peptidase P (ClpP) [35], decreases cell proliferation. Similarly, mtUPR promotes the growth and survival of cancer cells by ensuring mitochondrial function in the presence of mitochondrial stress related to cancer cell physiology [24]. We observed that mtUPR triggered by hydrogen exerted similar effects in the responders (Figure 1 and Figure 5). In addition, hydrogen was able to alleviate rotenone-induced suppression of cellular proliferation in the responders (Figure 6). In accordance with our observation of mtUPR-mediated enhancement of cancer cell growth by hydrogen, similar mtUPR-mediated effects have been reported in studies irrelevant to hydrogen: (i) phosphorylation of eIf2α [36,37], (ii) increase in ATF4 [37], (iii) activation of ATF5 [38,39], and (iv) activation of HSP60 [40] enhance cancer growth in mammals.

Elimination of ROS by molecular hydrogen has been repeatedly reported [6], but a direct radical-scavenging effect of hydrogen has been challenged [5,41]. First, a reaction rate constant between the hydroxyl radical and hydrogen is too low compared to those of other radical scavengers [41]. Second, the amount of hydrogen taken into the animal and human bodies is too low to exert efficient elimination of ROS. Third, the dwell time of hydrogen inside our bodies is too short. Fourth, the relatively small amount of orally administered hydrogen compared to the large amount of hydrogen that is constantly produced by intestinal microbiota is unlikely to efficiently scavenge enough ROS to make it biologically significant. In contrast to radical scavenging, we reported that inhalation of hydrogen gas mildly increased urinary excretion of the oxidative marker, 8-hydroxy-2’-deoxyguanine, in patients with Parkinson’s disease [42]. Similarly, hydrogen increased mitochondrial superoxide production in SH-SY5Y cells [14]. Mitochondria-derived ROS activates the Keap1-Nrf2 antioxidant signaling pathway [43]. Hydrogen has been repeatedly reported to activate the Keap1-Nrf2 pathway in both animal models [10,11,12,15] and cultured cells [8,13,14], indicating that hydrogen exerts its effects by enhancing the antioxidant signaling pathway. We showed that hydrogen did not increase the growth of SH-SY5Y cells (a non-responder). As hydrogen activates the Nrf2 signaling pathway in SH-SY5Y cells [14], the Nrf2 pathway is unlikely to be involved in the hydrogen-mediated induction of cell proliferation. The basal cellular ROS levels were similar between the responders and non-responders (Figure 2). In addition, hydrogen similarly decreased the average cellular ROS levels in both the responders and non-responders (Appendix A). Thus, the cell proliferation effect of hydrogen is unlikely to be mediated by the suppression of cellular ROS levels. However, we observed that inductions of mtUPR were different between the responders and non-responders. Mitochondria is a key organelle that plays an essential role in cellular metabolism, inflammation, and neurodegeneration [44]. Reduced mitochondrial membrane potential triggers mitophagy, which is likely compromised in Parkinson’s disease [45]. The mtUPR pathway is predicted to be activated before mitophagic degradation of compromised mitochondria [46]. We indeed reported the beneficial effects of hydrogen on Parkinson’s disease in rodents [47,48] and humans [49]. Even a small amount of hydrogen was likely to have alleviated the progression of Parkinson’s disease [50]. The effects of hydrogen on Parkinson’s disease may at least partially be explained by mtUPR-mediated rejuvenation of compromised mitochondria, such as what we observed in the responders where hydrogen rescued the effects of rotenone.

A recent report indicates that a high NAD^+^/NADH ratio drives cell proliferation [51]. To increase the NAD^+^/NADH ratio, cancer cells block pyruvate dehydrogenase (PDH), which converts NAD^+^ to NADH. As the block of PDH restricts an energy source for the TCA cycle and subsequent mitochondrial ETC, cancer cells become more dependent on glycolysis, which is known as the Warburg effect. Other strategies to increase the NAD^+^/NADH ratio are to increase electron flux from NADH to mitochondrial ETC and to activate lactate dehydrogenase. Hydrogen may exploit these mechanisms to induce cell proliferation. The Warburg effect is more prominent in undifferentiated cancer cells [52]. We showed that hydrogen enhanced cell proliferation of cancer cells with high mitochondrial ETC activities. Hydrogen may thus enhance the proliferation of differentiated cancer cells more than undifferentiated cancer cells. Interestingly, hydrogen induces differentiation of glioblastoma, the most undifferentiated form of glioma, and suppresses its growth in an animal model, although the involvement of mtUPR was not analyzed [53].

Considering that cancer sciences constitute one of the largest research fields in biomedical sciences [54,55], the paucity of reports on the effects of hydrogen on cancer may be partly due to masking of the favorable effects by the tumor-proliferating effects in a subset of cancer cells that can provoke mtUPR in response to hydrogen. Normal cells and cancer cells share similar metabolisms and signaling pathways. Therefore, a substance that protects normal cells against noxious stimuli and enhances the growth of normal cells should reasonably have a similar effect on cancer cells. However, hydrogen enhances cancer immunity, which may potentially circumvent the enhancement of cancer growth even if it exists [56,57]. Similarly, hydrogen alleviated adverse effects of chemotherapy [58,59] and radiotherapy [60]. Hydrogen suppressed cancer in several animal models [61,62,63,64] and in patients [65]. Although the underlying mechanisms remain unknown, hydrogen may suppress cancer by activating cancer immunity or by enhancing the survival of cancer-invaded tissues. In contrast to our studies, a higher concentration and a longer duration of hydrogen suppressed the growth of breast cancer cell lines (MCF-7, MDA-MB-453, and TUBO) [66] and induced apoptosis in the A549 lung cancer cell line [67,68,69], which was a responder in our studies. Although publication bias should exist, the lack of statistically significant effects of hydrogen on cancer in mouse models has also been reported [58,70]. Extensive and scrutinized pre-clinical and clinical studies are required to elucidate the favorable and unfavorable effects of hydrogen on various types of cancer including dose responses.

## 4. Materials and Methods

### 4.1. Cell Culture

A549, SH-SY5Y, HepG2, HT1080, PC3, HCT116, and HeLa cells are derived from human cancer cells and were obtained from Riken BRC. Medium for HepG2, HT1080, HCT116, and HeLa cells was “DMEM, high glucose” (11965126, Gibco, Waltham, MA, USA) with 10% fetal bovine serum (FBS, Gibco), 1% penicillin-streptomycin (Gibco), 20 mM HEPES-NaOH pH 7.4. Medium for A549 cells was “MEM” (11095080, Gibco) with 10% FBS, 1% Penicillin-Streptomycin, and 20 mM HEPES-NaOH pH 7.4. Medium for SH-SY5Y cells was “DMEM, high glucose, no glutamine” (11960, Gibco) with 10% FBS, 1% PS, 2 mM L-glutamine (Gibco), and 1% non-essential amino acids for MEM Eagle (MP Biomedical, Irvine, CA, USA). Cells were cultured in a 37 °C humidified incubator with 5% CO_2_.

### 4.2. Cell Culture with Hydrogen Gas

Cells were cultured under hydrogen as previously described [16]. Briefly, cells were cultured in a 6-well culture dish in a 560 mL closed plastic box humidified with a small amount of water at the base of the box at 37 °C. In the hydrogen group, 100% hydrogen gas (3 mL/min or 6 mL/min) was mixed with CO_2_-added air (5% CO_2_ and 95% air, 60 mL/min) to make 5% or 10% hydrogen gas. In the control group, 6 mL/min of 100% nitrogen gas was used instead of 100% hydrogen gas. We confirmed that the hydrogen concentrations in the medium became ~25 µM with 5% hydrogen gas and ~50 µM with 10% hydrogen gas at 20 min after the initiation of hydrogen (Figure S1i in [16]).

### 4.3. Cellular Proliferation Analysis

Cellular proliferation was assessed using the BrdU assay (Roche, Basel, Switzerland). Cells were seeded in a 96-well plate at 1 × 10^5^ cells per well. After 48 h incubation, the BrdU assay was performed according to the manufacturer’s protocols.

### 4.4. Cellular ROS Measurement

Cells were incubated under 10% hydrogen gas or 10% nitrogen gas for evaluation of cellular ROS production. The cells were trypsinized and incubated with 20 µM DCFDA (Abcam, Cambridge, UK) dissolved in DMEM for 30 min according to supplier’s protocols. DCFDA fluorescence of each cell was measured using FACS Calibur (BD Bioscience, Franklin Lakes, NJ, USA).

### 4.5. Mitochondrial Function Assay

MitoTracker Green (50 nM, Thermo Fisher Scientific, Waltham, MA, USA) or MitoSOX (5 µM, Thermo Fisher Scientific) was dissolved in Hanks’ Balanced Salt Solution (HBSS). Tetramethylrhodamine (200 nM, TMRM, Thermo Fisher Scientific) was dissolved in DMEM medium. The cells were stained for 30 min at 37 °C in a CO_2_ incubator and were collected by 0.25% trypsin and 0.1% EDTA. The fluorescence of each cell was measured using FACS Calibur (BD Bioscience).

### 4.6. Oxygen Consumption Rate Analysis

Cells were seeded at 1 × 10^4^ per well of the Seahorse XFp Cell Culture Miniplate (Agilent, Santa Clara, CA, USA) one day prior to measurement. Cells were incubated with the Seahorse XF Base Medium (without phenol red) (Agilent) supplemented with 10 mM glucose, 1 mM sodium pyruvate, 2 mM L-glutamine, and 5 mM HEPES-NaOH pH 7.4 for 1 h at 37 °C in a CO_2_-free incubator. The oxygen consumption rate (OCR) of the cells was measured by the Seahorse XFp Extracellular Flux Analyzer (Agilent) according to the manufacturer’s protocols. Oligomycin (1 µM), FCCP (0.5 µM for A549 cells or 1.0 µM for the other six cell lines), and rotenone/antimycin A (0.5 µM each) were serially injected into the well, and the OCR was measured for one min three times at each step. The OCR was analyzed by Seahorse Wave software (Agilent).

### 4.7. Cellular Doubling Time Analysis

Cellular doubling time was assessed using the IncuCyte ZOOM live cell imaging system (Essen BioScience, Göttingen, Germany). The system automatically takes images of cells in real time and measures cellular confluence per well by phase-contrast imaging at each time point. Cells were seeded in a 12-well plate at 10–15% confluency. Once the cells were seeded with fresh medium, the plates were placed into the IncuCyte ZOOM apparatus and images of the proliferating cells were recorded every 2 h for 72 h. The percentage confluence was measured by IncuCyte ZOOM software. Doubling time was calculated using an exponential growth phase from 20% to 80% confluence. 

### 4.8. Preparation of Cell Lysates and Western Blotting

Cells, after hydrogen gas incubation, were washed with PBS and harvested with PLC buffer (50 mM HEPES pH 7.0, 150 mM NaCl, 10% glycerol, 1% TritonX-100, 1.5 mM MgCl_2_, 1 mM EGTA, 100 mM NaF, and 10 mM sodium pyrophosphate) containing 1 μg/mL aprotinin, 1 μg/mL leupeptin, 1 μg/mL pepstatin A, and Phosphatase Inhibitor Cocktail (PhosSTOP, Roche). The lysates were incubated at 4 °C for 20 min and then centrifuged at 18,000 × *g* at 4 °C for 15 min. The supernatant was denatured for 5 min at 95 °C in Laemmli buffer, separated by electrophoresis on a 10% or 12% SDS-polyacrylamide gel, and blotted onto a polyvinylidene fluoride membrane (Immobilon-P, Millipore, Billerica, MA, USA). Membranes were washed in Tris-buffered saline containing 0.05% Tween 20 (TBS-T) and incubated for 1 h at room temperature in TBS-T with 5% skim milk or 5% bovine serum albumin (BSA). The membranes were reacted overnight at 4 °C with specific antibodies shown in Appendix A. The membranes were washed with TBS-T and incubated with secondary goat anti-mouse IgG (1:2000, LNA9310V/AG, GE Healthcare, Chicago, IL, USA) or anti-rabbit IgG (1:2000, LNA9340V/AE, GE Healthcare) antibody conjugated to horseradish peroxidase (HRP) for 1 h at room temperature. The bound antibodies were detected with Amersham ECL Western blotting detection reagents (GE Healthcare) and ImageQuant LAS 4000 mini (GE Healthcare). Signal intensity was quantified by ImageQuant TL software (GE Healthcare).

### 4.9. Treatment of Cells with Rotenone

Cells were seeded on a 96-well plate and treated with 1 nM to 10 µM rotenone for 48 h, and cell proliferation was assessed by BrdU assay. The rotenone concentration that was needed to induce approximately 20% cell death (LD20) was determined for each cell line (Appendix A). Each cell line was treated with rotenone at LD20 and was cultured in 5% hydrogen, 10% hydrogen, or 10% nitrogen for 48 h. After 48 h, a BrdU assay was performed to measure cell proliferation.

### 4.10. Gene Set Enrichment Analysis (GSEA) of 28 Microarray Datasets of the Seven Cell Lines

GSEA of the seven cell lines was performed using the public database. For example, 14 expression microarray datasets of normal HeLa cells were obtained from the Gene Expression Omnibus (GEO) database. The 14 expression profiles were individually subjected to principal component analysis (PCA) using R software. Then, four datasets with the smallest Euclidean distances to the center of gravity of the 14 datasets were selected (Appendix A). Four datasets were similarly selected for each of the other six cell lines. Differential expression profiles between sixteen datasets from the responders and twelve datasets from the non-responders were analyzed by GSEA software version 4.1.0 with the MsigDB 7.1 database.

### 4.11. Statistical Analyses

Statistical analyses were performed using Prism 8.4.3 (GraphPad Software, San Diego, CA, USA). Student’s *t*-test, one-way ANOVA, or two-way repeated-measures ANOVA were applied. For comparing every mean with every other mean after one-way and two-way repeated-measures ANOVA, Tukey’s and Sidak’s multiple comparison tests were applied, respectively. For comparing every mean with every other mean after two-way ANOVA, Sidak’s multiple comparison test was applied. For comparing a control mean with every other mean after one-way ANOVA and two-way repeated-measures ANOVA, Dunnett’s multiple comparison test was applied. *p*-values less than 0.05 were determined to be statistically significant.

## 5. Conclusions

We showed that hydrogen enhances cell proliferation in a subset of cancer cells that have high mitochondrial ETC activities and a readily inducible mtUPR pathway. The fact that hydrogen rescued mitochondrial stress suggests mitochondria and the associated mtUPR pathway are likely to play critical roles in determining the cellular responses to hydrogen. Our finding implies that stress relief by hydrogen may benefit cancer growth as well as normal or diseased cells via mitochondria.

## Figures and Tables

**Figure 1 ijms-23-02888-f001:**
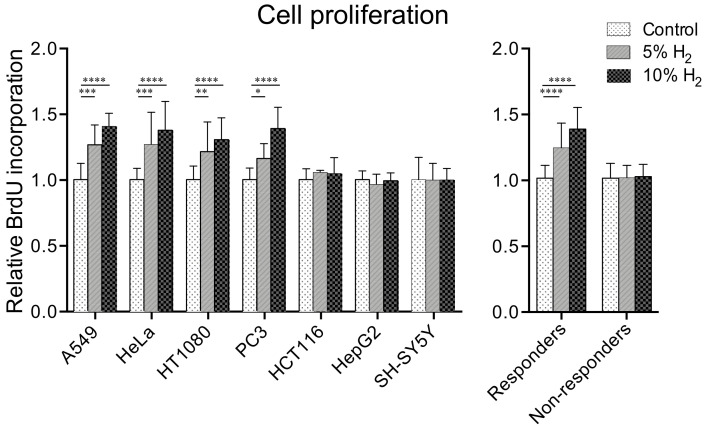
Molecular hydrogen enhances cell proliferation in four of seven cancer cell lines in a dose-dependent manner. The amount of incorporated BrdU into the cells incubated under 5 and 10% hydrogen gas was normalized for that of 0% hydrogen gas (control). Nitrogen gas was added at 10, 5, and 0% to substitute for 0, 5, and 10% hydrogen gas, respectively. Mean and SD are indicated (*n* = 8 culture dishes). * *p* < 0.05, ** *p* < 0.01, *** *p* < 0.001, and **** *p* < 0.0001 by two-way repeated-measures ANOVA followed by Dunnett’s multiple comparison test compared to control.

**Figure 2 ijms-23-02888-f002:**
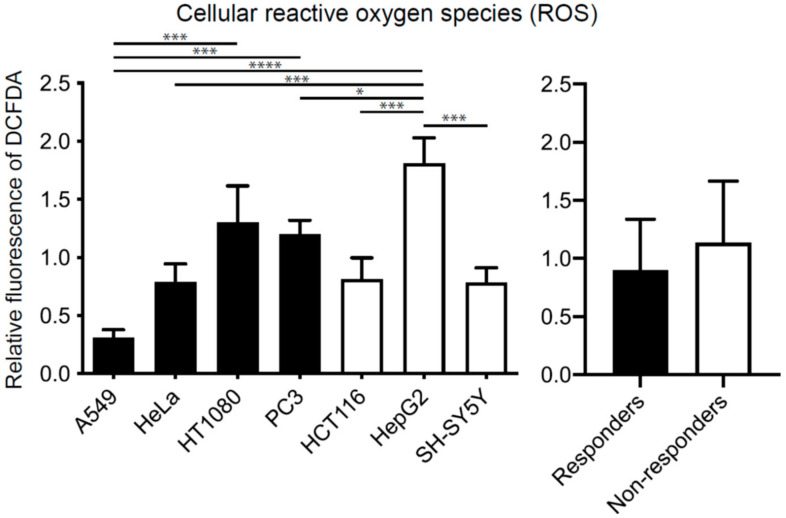
Basal concentrations of cellular reactive oxygen species (ROS) in seven cell lines. The fluorescence of DCFDA was normalized for the global average of the seven cell lines. Mean and SD are indicated (*n* = 3 culture dishes). * *p* < 0.05, *** *p* < 0.001 and **** *p* < 0.0001 by one-way ANOVA followed by Tukey’s multiple comparison test. No statistical difference (*p* = 0.28) by Student’s *t*-test between the responders and non-responders.

**Figure 3 ijms-23-02888-f003:**
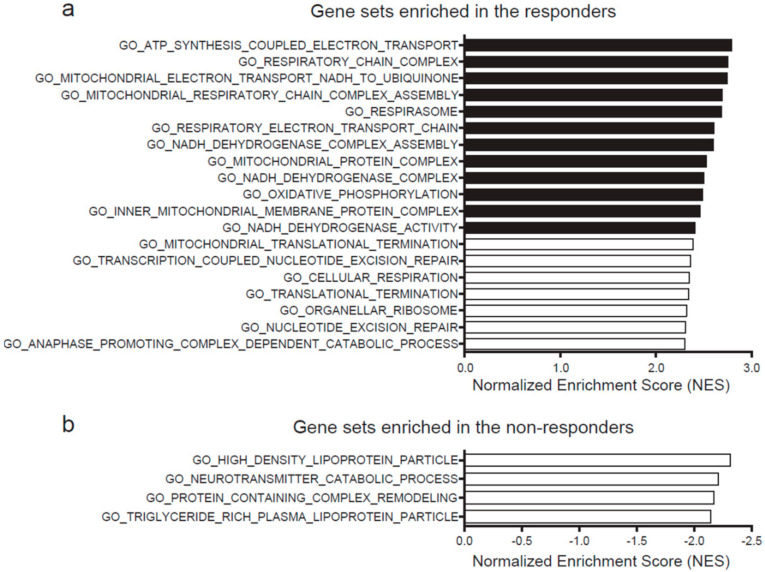
Genes sets enriched in the responders (**a**) and the non-responders (**b**). Gene sets with familywise error rate (FWER) *p*-value < 0.05 are indicated in descending order of normalized enrichment score (NES) of GSEA.

**Figure 4 ijms-23-02888-f004:**
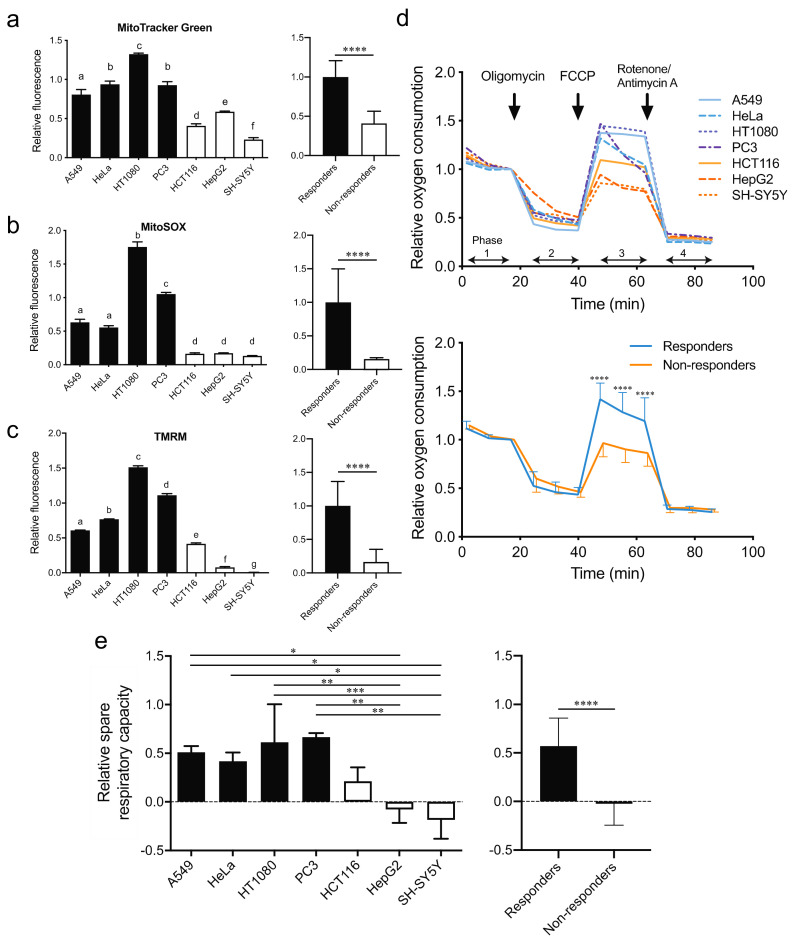
Mitochondrial functions of seven cell lines. Flow cytometric analyses of relative fluorescence signals for MitoTracker Green representing the amount of mitochondria (**a**), MitoSOX representing the amount of mitochondrial superoxide (**b**), and TMRM representing the mitochondrial membrane potential (**c**). Mean and SD are indicated (*n* = 3 culture dishes). Single letter labels indicate *p* < 0.05 by one-way ANOVA followed by Tukey’s multiple comparison test. Values in the same label are not statistically different from each other (e.g., d is different from a–c, but not from another d). **** *p* < 0.0001 by unpaired Student’s *t*-test between the responders and non-responders. (**d**) Relative oxygen consumption rates of each cell line that were measured by an extracellular flux analyzer. Oxygen consumption was normalized for that of untreated cells (Phase 1). Arrows indicate when the inhibitors were injected. Mean and SD indicated for the responders and non-responders (**** *p* < 0.0001 by two-way repeated-measures ANOVA followed by Sidak’s multiple comparison test). (**e**) Relative spare respiratory capacity of oxygen consumption of each cell line. The relative capacity was calculated by (Phase 3 − Phase 4)/(Phase 1 − Phase 4) − 1. Mean and SD are indicated (*n* = 3 culture dishes). * *p* < 0.05, ** *p* < 0.01, and *** *p* < 0.001 by one-way ANOVA followed by Tukey’s multiple comparison test. **** *p* < 0.0001 by unpaired Student’s *t*-test between the responders and non-responders.

**Figure 5 ijms-23-02888-f005:**
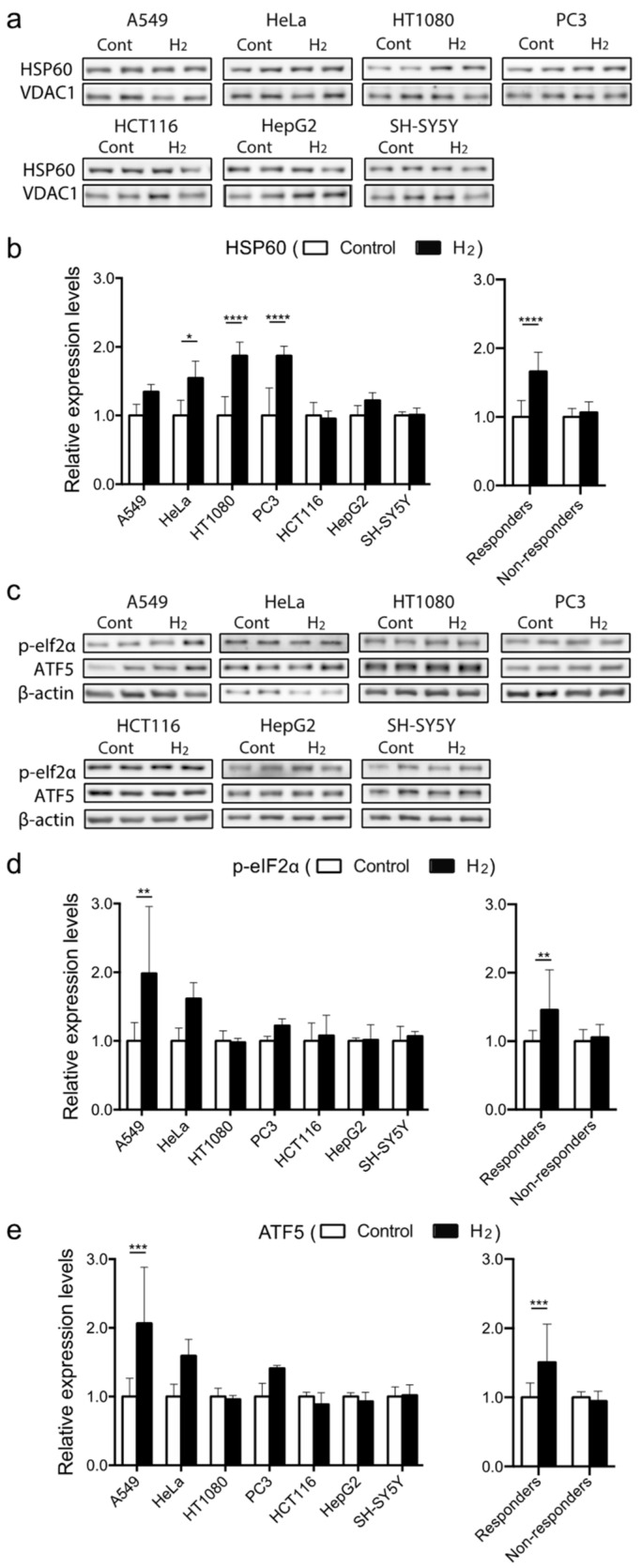
Expression of mtUPR-related molecules in seven cell lines under 10% hydrogen gas. Western blotting for HSP60 and VDAC1 under 10% hydrogen gas for 3 h. (**a**,**b**) The expression levels were normalized for that under 10% nitrogen gas as a control. Western blotting for phosphorylated eIf2α and ATF5 under 10% hydrogen gas for 2 h. (**c**–**e**) β actin was used as a loading control. The expression levels were normalized for that under 10% nitrogen gas as a control. Mean and SD are indicated (*n* = 3 culture dishes). * *p* < 0.05, ** *p* < 0.01, *** *p* < 0.001, and **** *p* < 0.0001 by two-way repeated-measures followed by Sidak’s multiple comparison test for each cell line, and by unpaired Student’s *t*-test between the responders and non-responders.

**Figure 6 ijms-23-02888-f006:**
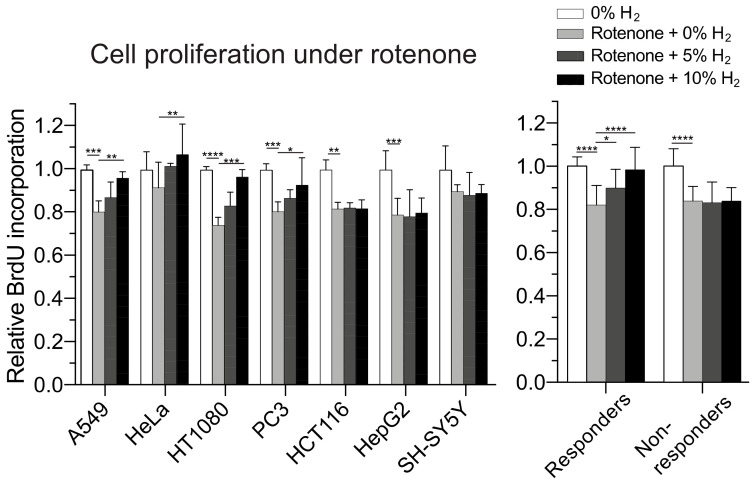
Effects of hydrogen on cell proliferation under mitochondrial stress. Relative amount of incorporated BrdU in the cells incubated under 0, 5, and 10% hydrogen gas with an inhibitor of mitochondrial electron transport chain complex I, rotenone. Nitrogen gas was added at 10, 5, and 0% to substitute for 0, 5, and 10% hydrogen gas, respectively. BrdU incorporation was normalized for that of cells without rotenone or hydrogen. Mean and SD are indicated *(n* = 4 culture dishes). * *p* < 0.05, ** *p* < 0.01, *** *p* < 0.001, and **** *p* < 0.0001 by two-way repeated-measures ANOVA followed by Dunnett’s multiple comparison test compared to rotenone with 0% hydrogen gas as a control.

## Data Availability

Additional data related to this paper are available upon request to the authors. The accession numbers of GEO microarray datasets downloaded for gene set enrichment analyses are shown on Appendix A. The whole exome sequencing data were deposited at DDBJ with an accession number DRA011366.

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
