# Peer review of "Molecular Hydrogen Enhances Proliferation of Cancer Cells That Exhibit Potent Mitochondrial Unfolded Protein Response"

_ijms, 2022, doi:10.3390/ijms23052888_

Round 1
Reviewer 1 Report
Reviewer comments and suggestions
The current study demonstrated the effects of molecular hydrogen on cancer cells and explained the underlying molecular mechanisms. The authors found that hydrogen enhances the proliferation of four out of seven human cancer cell lines (the responders). Expression profiling of the seven cells revealed that the responders have higher gene expression of mitochondrial electron transport chain (ETC) molecules including higher mitochondrial mass, higher mitochondrial superoxide, higher mitochondrial membrane potential, and higher mitochondrial spare respiratory capacity than the non-responders. It was suggested by the authors that In the responders, hydrogen provoked mitochondrial unfolded protein response (mtUPR) in the responder and concluded by an inhibitor of mitochondrial ETC complex I.
Below are the comments for this paper to be incorporated in the revised version of the manuscript.
- Line 38, explore the reference (4-6)
- Line 44 what would be the inference of this protective or deleterious (reference 16)
- Figure 6 demonstrated that other cells also showed effect how the authors explain this figure
- Line 203 is this required reference 29, I do not think the line needed
- Line 208 please mention brief about SNV here
- Line 240 mitohormesis used by the authors, please explore here at least 2-3 lines
- Line 247-248 what does it indicate
- Line 262-263 is not important line, please delete here and line 264-266 the authors need to mention the exact cause of using the reference, just highlighting is not important
- Line 271-272 what would be the reason for this
- Please rewrite the sentence line 302-303
- Line 306 how did the author explain these studies
- Delete the lines 314-317 and a typo error is seen in line 426, modify it
Reviewer 2 Report
In this manuscript, Hasegawa and colleagues report that molecular hydrogen promotes proliferation of cancer cells. By testing the response to H2 in different cancer cell lines, the authors could divide them into two groups: responder and not-responder. Based on these results, they then showed that responder cells have an increased activity of the mitochondrial electron transport chain (ETC), which correlates with a higher expression of mitochondrial genes. The authors finally present a set of experiments demonstrating that molecular hydrogen induces mitochondrial unfolded protein response (mtUPR) in hydrogen-responsive cells.
Overall, this manuscript is well written, simple, and perfectly intelligible. I think it could be of interest to the wide audience of the International Journal of Molecular sciences.
Author Response
Comments by Reviewer #2
We cordially appreciate scrutinizing comments by the reviewer.
We appreciate productive and encouraging comments by reviewer #2.